# Random-Noise Denoising and Clutter Elimination of Human Respiration Movements Based on an Improved Time Window Selection Algorithm Using Wavelet Transform

**DOI:** 10.3390/s19010095

**Published:** 2018-12-28

**Authors:** Farnaz Mahmoudi Shikhsarmast, Tingting Lyu, Xiaolin Liang, Hao Zhang, Thomas Aaron Gulliver

**Affiliations:** 1Department of Electronic Engineering, Ocean University of China, Qing Dao 266100, China; mahmoudi.farnaz@gmail.com; 2Science and Technology on Electronic Test & Measurement Laboratory, The 41st Research Institute of CETC, Qingdao 266555, China; iamxiaolin2016@126.com; 3Department of Electrical Computer Engineering, University of Victoria, PO Box 1700, STN CSC, Victoria, BC V8W 2Y2, Canada; agullive@ece.uvic.ca

**Keywords:** vital sign, ultra-wideband impulse radar, wavelet packet decomposition

## Abstract

This paper considers vital signs (VS) such as respiration movement detection of human subjects using an impulse ultra-wideband (UWB) through-wall radar with an improved sensing algorithm for random-noise de-noising and clutter elimination. One filter is used to improve the signal-to-noise ratio (SNR) of these VS signals. Using the wavelet packet decomposition, the standard deviation based spectral kurtosis is employed to analyze the signal characteristics to provide the distance estimate between the radar and human subject. The data size is reduced based on a defined region of interest (ROI), and this improves the system efficiency. The respiration frequency is estimated using a multiple time window selection algorithm. Experimental results are presented which illustrate the efficacy and reliability of this method. The proposed method is shown to provide better VS estimation than existing techniques in the literature.

## 1. Introduction

Noncontact measurement of vital signs (VS) has been the subject of significant research in recent years. It is used in applications such as health monitoring [1,2,3,4,5], heart rate variability analysis [6], monitoring chronic heart failure (CHF) patients [7], cancer radiotherapy [8], search and rescue [9], and animal health care [10,11,12,13]. Continuous wave (CW) radar has been used extensively for VS detection [14,15,16,17,18,19,20,21]. Many detection techniques have been proposed in the literature [22,23,24,25,26,27,28,29,30,31,32,33,34,35,36,37,38,39,40,41]. Ultra-wideband (UWB) impulse radar is one of the most effective methods for VS detection due to the high resolution and penetrability and low-power radiation [11,12,13,14,15].

A variable time window algorithm was used to estimate the human heart rate in [4]. Human cardiac and respiratory movements have been analysed using the fast Fourier transform (FFT) and Hilbert–Huang transform (HHT) [22,23]. To avoid the codomain restriction in arctangent demodulation (AD), an extended differentiate and cross-multiply (DACM) method was proposed in [24]. In [25], respiration-like clutter was suppressed using an adaptive cancellation method. A phase-based method is proposed to detect the heart rate based on the UWB impulse Doppler radar [26]. A short-time Fourier transform (STFT) was used for VS detection in [27]. However, these methods cannot accurately estimate the frequencies of VS signals due to the presence of harmonics. Further, STFT performance is sensitive to the window length. Ensemble empirical mode decomposition (EEMD) and a frequency window were used to remove clutter and harmonics in [28], but this increases the receiver complexity. In [28], singular value decomposition (SVD) was employed to detect the period of human respiration signals in very low signal to noise and clutter ratio (SNCR) environments. An adaptive Kalman filter was developed to extract respiration signals from UWB radar data in [29]. However, incorrect results were obtained when no subject was present. A constant false alarm rate (CFAR) technique was used in [32] to remove clutter and improve the SNR of VS signals. Static clutter was removed using linear trend subtraction (LTS) in [34]. In [36], a higher order cumulant (HOC) method was used to suppress noise based on the fact that the noise can be approximated as Gaussian. In [37], an EEMD method was presented to analyse human heart rate variations. An extended complex signal demodulation (CSD) technique was considered in [38] to eliminate the wrapped problem with the DACM algorithm. A state space (SS) method was proposed in [40] for VS detection over short distances. The usefulness of these algorithms is limited by the complexity of real VS signal environments. In particular, they are not effective for clutter removal, respiration signal extraction, and respiration and heart rate estimation in long distance and through-wall conditions. As a result, improved techniques are required for VS signal detection.

In this paper, a new method is presented to accurately estimate VS parameters even in low SNR conditions such as long range and through-wall. The time of arrival (TOA) is determined using the wavelet transform (WT) of the kurtosis and standard deviation (SD) of the received UWB signals. Further, a method to estimate the respiration frequency is presented. The performance of this method is compared with that of several well-known algorithms using data obtained with the UWB radar designed by the Key Laboratory of Electromagnetic Radiation and Sensing Technology, Institute of Electronics, Chinese Academy of Sciences. The contributions of the paper can be summarized as:The signal to noise and clutter ratio (SNCR) of the received UWB pulses is improved using an improved filter. Based on the distance estimate, the region of interest (ROI) containing VS signals is defined to reduce the data size and improve the system efficiency. To obtain the respiration frequency more accurately, the time window selection algorithm is proposed to remove the random noise. 

The remainder of this paper is organized as follows. In Section 2, the VS signal model is introduced. The proposed detection method is presented in Section 3 and the performance of this method is compared with several well-known techniques in Section 4. Finally, Section 5 concludes the paper.

## 2. Vital Sign Model 

A model similar to that in [36] for UWB impulse radar signals is employed here. Slow time denotes the received pulses while fast time represents the range. Figure 1 illustrates the received pulses which have been modulated by the periodic human respiration movements [27]. 

The distance can be expressed as
(1)$$d\left(t\right)=d_{0}+A_{r}\sin\left(2\pi f_{r}t\right)$$
where $d_{0}$ is the distance between the center of the human thorax and the radar, *t* is the slow time, $A_{r}$ is the respiration amplitude, and $f_{r}$ is the respiration frequency. The received UWB impulse radar signal is
(2)$$\begin{array}{l} R\left(\tau \right)=\sum_{n=0}^{N-1}u\left(\tau -nT-\tau_{r}\right)*h_{r}\left(\tau \right)\\ \;\;\;\;\;\;\;\;\;+\sum_{n=0}^{N-1}\sum_{p=1,p\neq r}^{P}u\left(\tau -nT-\tau_{p}\right)*h_{p}\left(\tau \right)+a\left(\tau \right)+q\left(\tau \right)+g\left(\tau \right)+\omega \left(\tau \right) \end{array}$$
where ∗ denotes convolution, u(t) is the transmitted UWB pulse, *τ* is the fast time, T is the pulse period, *n* is the slow time index with N samples, $\tau_{r}$ is the time delay from UWB radar to the human subject, *r* denotes the response from human respiratory movements, $h_{r}\left(\tau \right)$ is the response from the human subject, *P* is the number of the static objects, $\tau_{p}$ is the time delay from UWB radar to other objects, $h_{p}\left(\tau \right)$ is the response from all other objects, a(τ) is the linear trend, w(τ) is additive white Gaussian noise (AWGN), q(τ) is nonstatic clutter, and g(τ) is unknown clutter. The time delay can be expressed as
(3)$$\tau_{\vartheta }=\tau_{0}+\tau_{r}\sin\left(2\pi f_{r}nT\right)$$
where $\tau_{0}=2d_{0}/v$ and $\tau_{r}=2A_{r}/v$, *v* is the light speed. The fast time period is $\delta_{T}$ with sampling interval $\delta_{R}=v\delta_{T}/2$. The received signal can be expressed as a M×N matrix R with elements
(4)R[m,n]=h[m,n]+c[m,n]+a[m,n]+w[m,n]+q[m,n]+g[m,n]
where *m* is the fast time index and *M* is the corresponding number of samples, h[m,n] is the received pulses from human subject in digital form, c[m,n] is the received pulses from static object in digital form, a[m,n] is the linear trend in digital form, w[m,n] is AWGN in digital form, q[m,n] is non-static clutter in digital form and g[m,n] is unknown clutter in digital form.

In a static environment, the ideal signal after clutter removal is
(5)$$R\left(\tau \right)=\sum_{n=0}^{N-1}u\left(\tau -nT-\tau_{r}\right)*h_{r}\left(\tau \right)$$

Taking the Fourier transform (FT) gives
(6)$$Y\left(m\delta_{T},f\right)=\int_{-\infty }^{+\infty }R\left(\tau \right)e^{-j2\pi ft}dt$$
and the two dimensional (2-D) FT gives
(7)$$Y\left(m\delta_{T},f\right)=\int_{-\infty }^{+\infty }Y\left(\upsilon ,f\right)e^{j2\pi \upsilon \tau }d\upsilon .$$
where
(8)$$Y\left(\upsilon ,f\right)=\int_{-\infty }^{+\infty }\int_{-\infty }^{+\infty }R\left(\tau \right)e^{-j2\pi ft}e^{-j2\pi \upsilon \tau }dtd\tau$$
(9)$$\begin{array}{l} Y\left(\upsilon ,f\right)=\int_{-\infty }^{+\infty }a_{v}U\left(\upsilon \right)e^{-j2\pi ft}e^{-j2\pi \upsilon \tau_{v}\left(t\right)}dt\\ \;\;\;\;\;\;\;\;\;\;\;=a_{v}U\left(\upsilon \right)e^{-j2\pi \upsilon \tau_{0}}\int_{-\infty }^{+\infty }e^{-j2\pi \upsilon m_{b}\sin\left(2\pi f_{r}t\right)}e^{-j2\pi \upsilon m_{h}\sin\left(2\pi f_{h}t\right)}e^{-j2\pi ft}dt \end{array}$$
U(υ) is the FT of a UWB pulse, *f* is the spectrums of the slow time, and υ is the spectrums of the fast time.

Using Bessel functions, Formula (12) can be expressed as
(10)$$Y\left(\upsilon ,f\right)=a_{v}U\left(\upsilon \right)e^{-j2\pi \upsilon \tau_{0}}\int_{-\infty }^{+\infty }\left(\sum_{k=-\infty }^{+\infty }J_{k}\left(\beta_{r}\upsilon \right)e^{-j2\pi kf_{r}t}\right)\left(\sum_{l=-\infty }^{+\infty }J_{l}\left(\beta_{h}\upsilon \right)e^{-j2\pi lf_{b}t}\right)e^{-j2\pi ft}dt$$

We have
(11)$$e^{-jz\sin\left(2\pi f_{0}t\right)}=\sum_{k=-\infty }^{+\infty }J_{k}\left(z\right)e^{-j2\pi kf_{0}t}$$
where $\beta_{r}=2\pi A_{r}$, and $\beta_{h}=2\pi A_{h}$, so then Formula (7) is given by
(12)$$Y\left(m\delta_{T},f\right)=a_{v}\sum_{k=-\infty }^{+\infty }\sum_{l=-\infty }^{+\infty }G_{kl}\left(\tau \right)\delta \left(f-kf_{r}-lf_{h}\right)$$
where
(13)$$G_{kl}\left(\tau \right)=\int_{-\infty }^{+\infty }U\left(\upsilon \right)J_{k}\left(\beta_{r}\upsilon \right)J_{l}\left(\beta_{h}\upsilon \right)e^{j2\pi \upsilon \left(\tau -\tau_{0}\right)}d\upsilon$$

When $m\delta_{T}=\tau_{0}$, the maximum of Formula (13) is obtained as
(14)$$C_{kl}=G_{kl}\left(\tau_{0}\right)=\int_{-\infty }^{+\infty }U\left(\upsilon \right)J_{k}\left(\beta_{r}\upsilon \right)J_{l}\left(\beta_{h}\upsilon \right)d\upsilon$$
(15)$$Y\left(\tau_{0},f\right)=a_{v}\sum_{k=-\infty }^{+\infty }\sum_{l=-\infty }^{+\infty }C_{kl}\delta \left(f-kf_{r}-lf_{h}\right)$$

The respiratory signal with l=0 is given by
(16)$$C_{k0}=\int_{-\infty }^{+\infty }S\left(\upsilon \right)J_{k}\left(\beta_{r}\upsilon \right)J_{0}\left(\beta_{h}\upsilon \right)d\upsilon$$

However, linear trend, non-static clutter, and other clutter exist in the received signals. This along with AWGN makes detection difficult, as can be seen by comparing Figure 2a,b which show a respiration signal without and with AWGN [30], respectively.

## 3. VS Detection Algorithm

A flowchart of the proposed detection method is given in Figure 3 and the steps are described below. Five healthy volunteers from the Key Laboratory of Electromagnetic Radiation and Sensing Technology, Institute of Electronics, participated in this research. All participants consented to participate and were informed of the associated risks. The experiments were approved by both Ocean University of China and the Chinese Academy of Sciences and were performed in accordance with the relevant international guidelines and regulations.

### 3.1. Clutter Suppression

VS signals are typically corrupted by significant static clutter which can be estimated as
(17)$$J=\frac{1}{M\times N}\sum_{m=1}^{M}\sum_{n=1}^{N}R\left[m,n\right]$$
and the signal after cancellation is
(18)Ω[m,n]=R[m,n]−J

The LTS algorithm can be used to remove the linear trend term expressed as [27,30]
(19)$$W=\Omega^{T}-X{\left(X^{T}X\right)}^{-1}X^{T}\Omega^{T}$$
where $X=\left[x_{1},x_{2}\right]$, $x_{1}={\left[0,1,\ldots ,N-1\right]}^{T}$, $x_{2}={\left[1,1,\ldots ,1\right]}_{N}^{T}$, and T denotes the matrix transpose.

### 3.2. SNR Improvement

The received signal depends on the dielectric constant, humidity, and polarization of the electromagnetic wave, and estimating these values can be difficult [36]. Therefore, a bandpass filter is used rather than a matched filter. In this paper, two fifth-order Butterworth filters are employed, a low-pass filter with normalized cutoff frequency 0.1037 and a high-pass filter with normalized cut off frequency 0.0222. The filter output is
(20)$$\Lambda \left[m,n\right]=\alpha_{1}W\left[m,n\right]+\alpha_{2}W\left[m-1,n\right]+\ldots +\alpha_{6}W\left[m-5,n\right]-\beta_{2}W\left[m-1,n\right]-\ldots -\beta_{6}W\left[m-5,n\right]$$
where α and β are the coefficient vectors. A smoothing filter which averages seven values in slow time is used to improve SNR which gives
(21)$$\Phi \left[k,n\right]=\frac{\sum_{m=7\times \gamma }^{7\times \gamma +6}\Lambda \left[m,n\right]}{7}$$
where γ=1,…,⌊M/7⌋ and ⌊M/7⌋ is the largest integer less than M/7.

### 3.3. TOA Estimation

Gaussian noise is a major factor affecting VS signals. The spectral kurtosis can be used to extract non-Gaussian signals and their position in frequency [42,43] and has been employed in many applications [44,45,46,47,48]. An improved TOA estimation algorithm is presented here, which is based on the standard deviation (SD) and kurtosis. 

The kurtosis for each fast time index *m* in Φ is given by [44]
(22)$$K=\frac{E\left[{\left(\Phi \left[m,n\right]\right)}^{4}\right]}{{\left\{E\left[{\left(\Phi \left[m,n\right]\right)}^{2}\right]\right\}}^{2}}$$
where E[•] denotes expectation. The kurtosis is three for a Gaussian distribution [45] and the excess kurtosis is given by
(23)$$\tilde{K}=K-3$$

The excess kurtosis is considered in this paper, and this is given in Figure 4a for the data acquired with a volunteer located at a distance of 9 m from the radar outdoors. This shows that the difference in excess kurtosis between when a target is present and not present is small. Thus the SD is combined with the excess kurtosis. 

The SD is given by [44]
(24)$$SD=\sqrt{\frac{\sum_{n=1}^{N}{\left(\Phi_{m}-\mu \right)}^{2}}{N-1}}$$

And the kurtosis to SD (KSD) is defined as $\tilde{K}/SD$. Figure 4b shows the KSD for the radar data described above. This indicates that there is a significant difference in the KSD between when a subject is present and not present. Figure 4d shows that the KSD in the target area is approximately periodic. Figure 4d gives the FTT of the KSD in Figure 4d, which confirms the KSD periodicity. The KSD when a subject is not present is shown in Figure 4c.

The STFT [49,50,51] and wavelet transform (WT) [52,53,54] have been widely used to analyze VS signals. However, the STFT performance depends on the length which can be difficult to determine. As a consequence, the WT is considered here as it also provides the advantage of scalability in the frequency domain [55]. For a given time domain signal z(τ), the continuous WT is
(25)$$C=\frac{1}{\sqrt{a}}\int_{-\infty }^{\infty }z\left(\tau \right)\bar{\psi }\left(\frac{\tau -b}{a}\right)d\tau$$
where ψ((τ−b)/a) is the wavelet with scaling parameter a and translation parameter b, and ¯ denotes complex conjugate.

The Morlet wavelet is considered here as it is widely used because of its simple implementation and is given by
(26)$$\bar{\psi }\left(\tau \right)=e^{-\frac{\tau^{2}}{2}}\cos\left(5\tau \right)$$

The discrete WT is employed in this paper, which is
(27)$$D=\frac{1}{\sqrt{a}}\sum_{n}z\left(n\right)\bar{\psi }\left(\frac{n-b}{a}\right)$$

The resulting time-frequency matrix when a target is present is shown in Figure 5a and when a target is not present in Figure 5b. The VS signal is indicated in the red region. The range between the radar receiver and target can be estimated as
(28)$$\hat{L}=v\times \hat{\tau }/2$$
where $\hat{\tau }$ is TOA estimate corresponding to the maximum value in the matrix.

### 3.4. Frequency Estimation

#### 3.4.1. Data Reduction

The index of the TOA estimate $\hat{\tau }$ in Φ can be expressed as
(29)$$ℑ=\hat{\tau }/2\delta_{T}$$

The human respiration frequency is usually between 0.2 and 0.4 Hz with amplitude 0.5 to 1.5 cm [56]. Thus, ε∈[ℑ−10,ℑ+10], which has a range of approximately 8 cm is considered as the region of interest (ROI) containing the respiration signal. ROI constants for all the human subjects, and it is independent of the height, weight, size of the person. Figure 6a shows a slow time signal in the ROI, while a slow time signal not in the ROI is given in Figure 6b. To further illustrate the differences, Figure 6c shows 10 randomly selected slow time signals in the ROI, Figure 6d shows 10 randomly selected slow time signals not in the ROI, and Figure 6e shows all the slow time signals in the ROI. These show that the transmitted radar signals have been modulated by the human respiration signal. Thus, only signals in the ROI are used to estimate the respiration frequency. In the radar system, 4096 × *N* samples are to reduce the amount data to be analyzed and improve performance.

#### 3.4.2. Noise Removal 

To estimate the respiration frequency more accurately, the wavelet packet decomposition of each slow time signal in the ROI is used [57,58]. For each slow time signal in Figure 6a where $\Phi_{\varepsilon }\left(n\right),\varepsilon \in \left[ℑ-10,ℑ+10\right]$, we have
(30)$$\Phi_{\varepsilon }\left(n\right)=\sum_{i=-\infty }^{\infty }a_{i}\psi^{*}\left(n-i\right)+\sum_{j=0}^{\infty }\sum_{i=-\infty }^{\infty }d_{j}2^{j/2}\psi \left(2^{j}n-i\right)$$
where
(31)$$a_{i}=\sum_{n=-\infty }^{\infty }\Phi_{\varepsilon }\left(n\right)\psi^{*}\left(n-i\right)$$
are the scaling coefficients and
(32)$$d_{j}=2^{j/2}\sum_{n=-\infty }^{\infty }\Phi_{\varepsilon }\left(n\right)\psi^{*}\left(2^{j}n-i\right)$$
are the wavelets. 

The corresponding Welch power spectrums of the individual wavelets are given in Figure 7 [59]. The *x*-axis is normalized frequency which is given by
(33)$$f_{n}=f/\left(\pi f_{s}\right)$$
where $f_{s}$ denotes the slow time sampling frequency. This figure shows that *d*_6_ is concentrated in the 0.1 to 0.5 Hz frequency range, while the other wavelets are concentrated in higher (>1 Hz) or lower (<0.1 Hz) frequencies. As a result, to reduce noise the VS signals are extracted using *d*_6_.

#### 3.4.3. Spectral Analysis

As mentioned above, the respiration period is between 2.5 and 5 s. As a result, it is challenging to estimate the respiration frequency accurately using a time window of less than 5 s. Further, the estimation accuracy is affected by inhomogeneous respiration, so the time window should be at least 3 to 6 periods [4]. An FFT is performed on *d*_6_ and the maximum value is the respiration frequency estimate. 

For a time window $T_{w}$, the resolution is
(34)$$\Delta f=1/T_{w}$$

Accurate estimation requires that
(35)$$\Delta f\ll f_{r}$$
so
(36)$$f_{r}=\rho \times \Delta f$$
where ρ is an integer chosen to satisfy Formula (35). 

Figure 8a shows the time windows used which are given by
(37)$$w_{\chi }=w_{1}+\ldots +w_{\chi -1}+\varsigma$$
where $w_{\chi }$ is the length of the *i*th window and *ς* denotes the increase in length. Increasing the window length improves the frequency resolution which is given by
(38)$$\Delta f_{\chi }=\frac{f_{s}}{w_{\chi }},\;\;\;\;\chi =1,2,\ldots$$
where $f_{s}$ is the sampling frequency, and *ς* satisfies [4]
(39)$$\varsigma =\rho \frac{f_{s}}{f_{r}}$$

This increase in window length results in an increase in the complexity of the radar system as multiple FFTs must be computed. Further, this approach cannot reduce the harmonics of the respiration signal [4] due to the different frequency spectrum lengths. 

To improve the frequency resolution and reduce the harmonics while keeping the complexity reasonable, another multiple time window technique is employed which is shown in Figure 8b. In this case, the time windows have the same length so the frequency resolution is
(40)$$\Delta f=\frac{f_{s}}{w_{i}},\;\;\;\;i=1,2,\ldots ,q$$
where $w_{i}$ is the length of the *i*th window, and *q* is the number of windows.

The radar system can only process data of length $2^{\phi }$. Therefore, the window length is chosen to be *w_i_* = 512 samples with an overlap of *G* = 256 in Figure 8b. Each radar measurement provides 1024 samples, so *q* = 3 sets of data are acquired for$\xi_{\varepsilon }$, ε∈[ℑ−10,ℑ+10]. Thus, the system parameters are *N* = 512, Δf=0.05 Hz, ρ=4 and ϕ=9.

A frequency window of 0.1 to 0.8 Hz was previously used to reduce the clutter and improve SNR. However, a window is not necessary with the proposed approach due to the defined ROI. As a result, only an FFT is performed on each signal
(41)$$\Omega \left[\delta \right]=\mathrm{FFT}\left\{\xi_{\lambda }\right\}$$

Cumulants are employed to remove harmonics and clutter
(42)$$H\left(i\right)=\sum_{j=1}^{21}\Omega_{j}\left(i\right)$$
The frequency is then estimated as
(43)$$f_{r}=H\left[\mu_{r}\right]$$
where $\mu_{r}$ corresponds to the index of the maximum value in Formula (37). 

## 4. Data Acquisition

### 4.1. UWB Impulse Radar

Figure 9a shows the UWB impulse radar used for data acquisition. It contains one transmitter and one receiver and is controlled by a wireless personal digital assistant. Table 1 gives the system parameters. The UWB pulses are transmitted with a 400 MHz center frequency and a 600 kHz pulse repetition frequency. The data are obtained for 124 ns time windows. *M* = 4096 samples are obtained in fast time and *N* = 512 pulses in slow time which requires 17.6 s. A combination of the equivalent-time [60] and real-time [61] sampling methods is employed which provides better performance than with only one method. Figure 9b shows the received signal matrix *R* obtained with one male volunteer outdoors at a distance of 9 m from the radar. The VS signal is not noticeable because of the large path loss due to the long-range and through-wall conditions. This indicates that VS signal detection in real environments is challenging.

### 4.2. Experimental Setup

The experiments were conducted at the Institute of Electronics, Chinese Academy of Sciences and the China National Fire Equipment Quality Supervision Centre. The experimental setups are illustrated in Figure 10a,b. The human subjects faced the radar breathing normally and kept still.

The first experiment was conducted outdoors at the Institute of Electronics as shown in Figure 10c at distances of 6 m, 9 m, 11 m and 14 m with two female (158 cm, 48 kg and 163 cm, 54 kg) and two male (178 cm, 84 kg and 182 cm, 76 kg) subjects. This environment includes vegetation with moves in the wind. The wall is 2.7 m high and more than 10 m wide, and is composed of three different materials including 30 cm of brick, 35 cm of concrete, and 35 cm of pine. The radar is located at a height of 1.5 m. The second experiment was conducted indoors at the China National Fire Equipment Quality Supervision Center as shown in Figure 10d at distances of 7 m, 10 m, 12 m and 15 m with one male subject (172 cm, 74 kg). The wall is 2.5 m in height and 3 m in width. The radar is located at a height of 1.3 m.

The third experiment was conducted indoors at the China National Fire Equipment Quality Supervision Center as shown in Figure 10e at distances of 7 m, 10 m and 12 m using an actuator of size 0.25 × 0.3 m^2^ to imitate human respiration. The actuator was on a desk 1.3 m above ground and moved at an amplitude of 3 mm and a frequency of 0.3333 Hz. In the fourth experiment, the actuator was on a desk 70 cm above ground at a distance of 6 m outdoors at the Institute of Electronics. In the fifth experiment, the actuator was on a desk 1.3-m above ground at azimuth angles of 30° and 60°with respect to the radar antenna indoors at a distance of 6 m as illustrated in Figure 10b.

## 5. Experimental Results

In this section, the performance of the proposed algorithm is compared with the FFT, constant false alarm rate (CFAR) [32], advanced [35], and multiple higher order cumulant (MHOC) [36] methods which are well-known in the literature. The clutter removal and SNR improvement are evaluated using data from the first experiment with a female subject (158 cm, 48 kg) located 9 m from the radar. Figure 11 shows the received signal for 18 s. The result after LTS is given in Figure 11a. This indicates that although the amplitude of the received signal is decreased, the VS signal is more pronounced. Figure 11b gives the result after filtering in fast time (range), and Figure 11c after filtering in slow time. This shows that the VS signal is more visible after filtering.

### 5.1. Vital Sign Estimation Outdoors

The TOA and frequency estimation performance are now examined using the data from experiment one with four subjects at different distances. The KSD, TOA, and frequency estimation are first obtained with one female subject (158 cm, 48 kg). Figure 12 presents the KSD for the four distances which shows that the KSD is larger in the ROI. The range results after WT decomposition of the KSD are shown in Figure 13. The range errors are 0.12 m at 6 m, 0.17 m at 9 m, 0.11 m at 11 m and 0.14 m at 14 m. The slow time signals in the ROI at the four distances are given in Figure 14. All indicate modulation by human respiration. Figure 15 presents the corresponding frequency estimation which gives values of 0.26 Hz at 6 m, 0.31 Hz at 9 m, 0.31 Hz at 11 m and 0.26 Hz at 14 m. Further, it indicates that the harmonics have been effectively suppressed.

The SNR of the VS signal can be expressed as [37]
(44)$$\mathrm{SNR}=20\log_{10}\left(\frac{\left|H\left[\mu_{r}\right]\right|}{\sum_{n=v_{1}}^{\mu_{r}-1}\left|H\left[n\right]\right|+\sum_{n=\mu_{r}+1}^{\nu_{2}}\left|H\left[n\right]\right|}\right)$$
where $\nu_{1}$ is the zero frequency index and $\nu_{2}$ is the index of $f_{s}/2$. Figure 16 gives the results for the CFAR method with subject III, and the corresponding advanced method (AM) results are shown in Figure 17. The red squares denote the estimates while the black ellipses denote the true values. These figures show that these methods cannot provide accurate range estimates, while the previous results indicate that the proposed method performs well even at a distance of 14 m. The frequency estimates and corresponding SNRs for the four subjects with the proposed algorithm are given in Table 2. Table 3 presents the results with subject I for four different algorithms. These tables show that the proposed method provides more accurate range and frequency estimates, and high SNRs.

### 5.2. VS Estimation Indoors

The data from the second experiment obtained indoors at the China National Fire Equipment Quality Supervision Center with a male subject (172 cm, 74 kg) is now considered. Figure 18 shows the KSD for distances of 7 m, 10 m, 12 m and 15 m. The range estimates after WT decomposition of the KSD are given in Figure 19. The corresponding errors are 0.04 m at 7 m, 0.05 m at 10 m, 0.08 m at 12 m and 0.07 m at 15 m. These results indicate that the range is estimated more accurately indoors, largely due to the fact that the wind causes movement in the environment.

Figure 20 shows the frequency estimation results and the estimates are 0.37 Hz at 7 m, 0.31 Hz at 10 m, 0.31 Hz at 12 m and 0.37 Hz at 15 m. The frequency estimates, SNRs and range errors for three methods are given in Table 4. The FFT method has the worst performance and cannot provide accurate frequency and range estimates. Adding a frequency window after the FFT can improve the SNR [62], but the range and frequency estimates are still poor, especially at long distances. Conversely, the proposed method has excellent performance at all distances. 

### 5.3. Actuator Signal Estimation

The KSD for the data from experiment three is shown in Figure 21. The corresponding range estimates obtained using WT decomposition are given in Figure 22, and the signals in the ROI are shown in Figure 23. Comparing Figure 14 and Figure 23, the modulation is more pronounced with the actuator than with human respiration. Figure 24 shows the frequency estimation and the estimates are 0.34 Hz at 7 m, 0.32 Hz at 10 m and 0.33 Hz at 12 m. The corresponding deviations are 0.66%, 0.33%, and 0.24%, respectively. The results for four algorithms are given in Table 5. Again the proposed method is the best. The detection results using the data from experiment four are shown in Figure 25. This shows that the proposed algorithm provides better range and frequency estimates compared to the other methods.

### 5.4. Estimation at Different Azimuth Angles

The influence of the azimuth angle between the subject and radar on the detection performance is now examined. The beam angle of the antenna in the radar system is 60°. In previous sections, the subject and actuator were directly in front of the radar so the azimuth angle was 0°. In this section, results are obtained for the data from experiment five with the actuator at a distance of 6 m and angles of 30 and 60. Figure 26 presents the KSD and range estimates, and the signals in the ROI are shown in Figure 27. Table 6 compares the results for the proposed algorithm with three other methods. This indicates that the proposed algorithm provides superior performance, particularly at a 60°angle. AM has a better frequency estimate at 30°but the range error is very high. 

## 6. Conclusions

In this paper, a new method for human respiration movement detection was presented based on an UWB impulse radar. The time of arrival (TOA) of the received UWB signals was estimated using a wavelet transform (WT), and the human respiration frequency was estimated using a time window technique. The performance of the proposed method was compared with several well-known algorithms under different indoor and outdoor conditions. The results obtained indicate that this technique can effectively suppress clutter and provides superior detection performance.

## Figures and Tables

**Figure 1 sensors-19-00095-f001:**
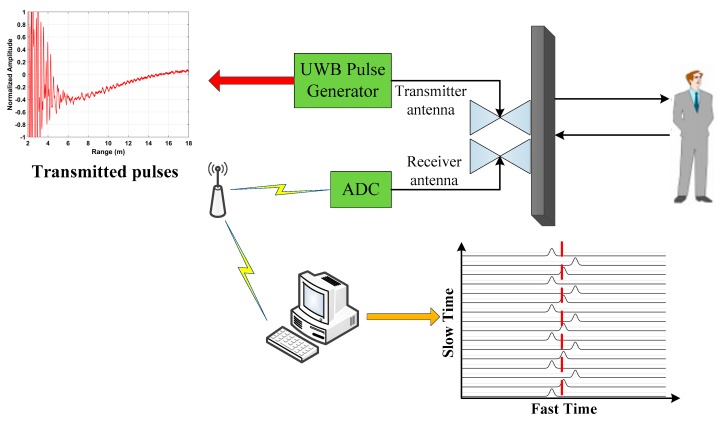
An illustration of the received radar pulses.

**Figure 2 sensors-19-00095-f002:**
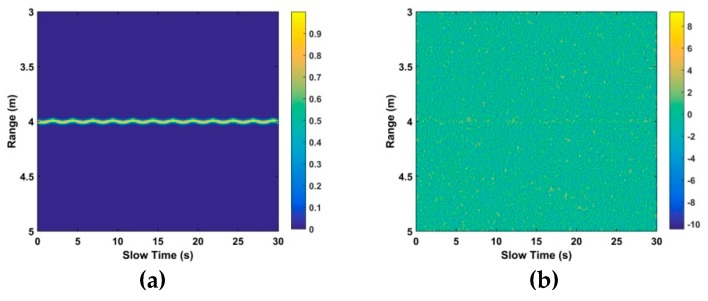
The time-range matrix with (**a**) only the respiration signal, and (**b**) the signal with additive white Gaussian noise (AWGN) at an SNR of −10 dB.

**Figure 3 sensors-19-00095-f003:**
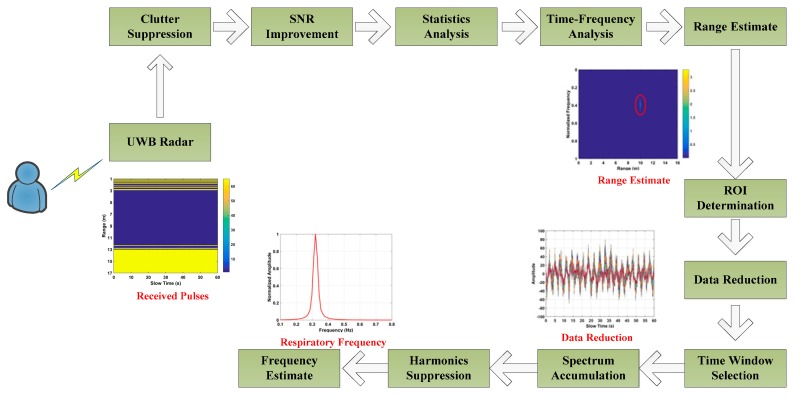
Flowchart of the proposed detection method.

**Figure 4 sensors-19-00095-f004:**
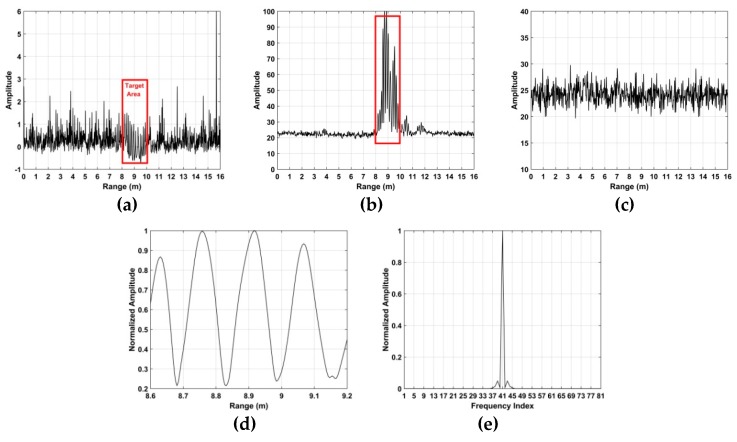
The (**a**) excess kurtosis, and (**b**) the kurtosis to SD (KSD) with a human subject, the (**c**) KSD without a human subject, (**d**) KSD in the target area, and (**e**) the spectrum of (**d**).

**Figure 5 sensors-19-00095-f005:**
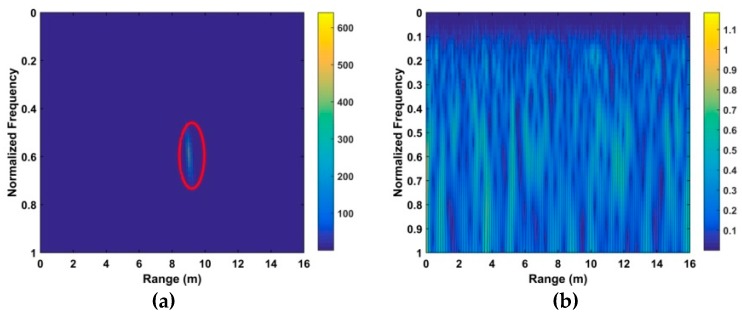
The time-frequency matrix using the wavelet transform (WT) decomposition (**a**) with a human subject at 9 m and (**b**) without a subject.

**Figure 6 sensors-19-00095-f006:**
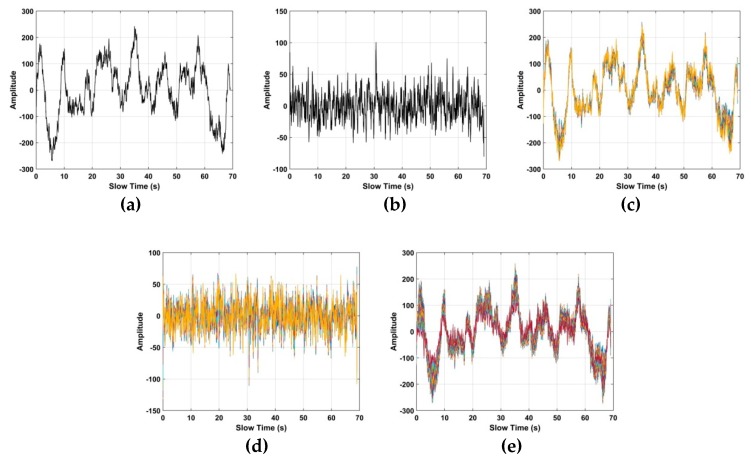
Signals in the time-frequency matrix (**a**) one in the region of interest (ROI), (**b**) one not in the ROI, (**c**) ten in the ROI, (**d**) ten not in the ROI and (**e**) all in the ROI.

**Figure 7 sensors-19-00095-f007:**
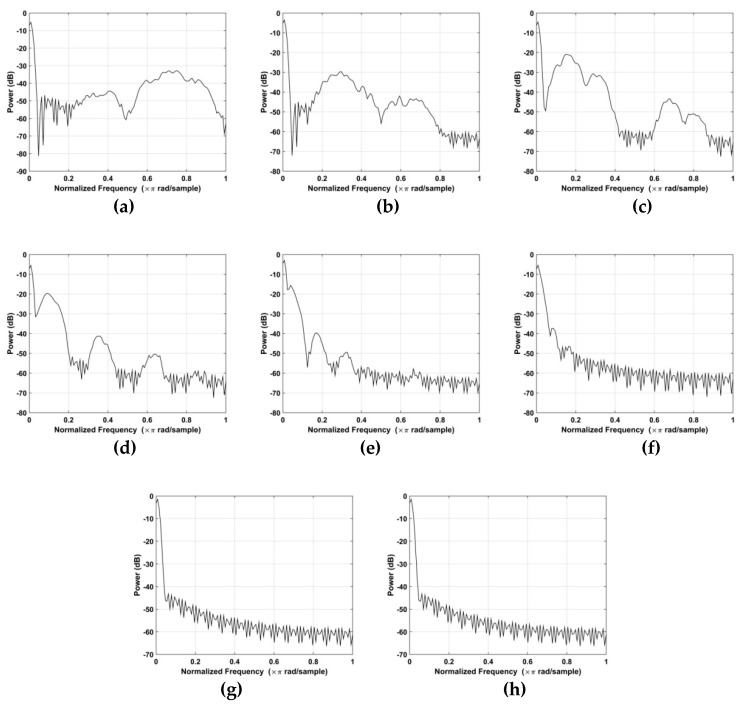
The Welch power spectrum of (**a**) *d*_1_, (**b**) *d*_2_, (**c**) *d*_3_, (**d**) *d*_4_, (**e**) *d*_5_, (**f**) *d*_6_, (**g**) *d*_7_, and (**h**) *d*_8_.

**Figure 8 sensors-19-00095-f008:**
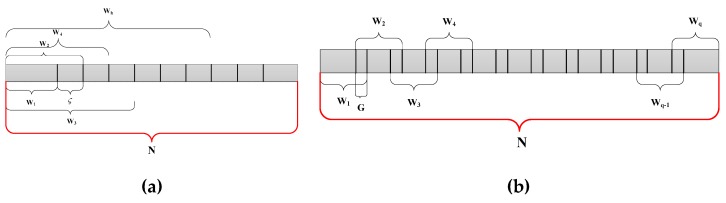
The time windows with (**a**) increasing length, and (**b**) fixed length.

**Figure 9 sensors-19-00095-f009:**
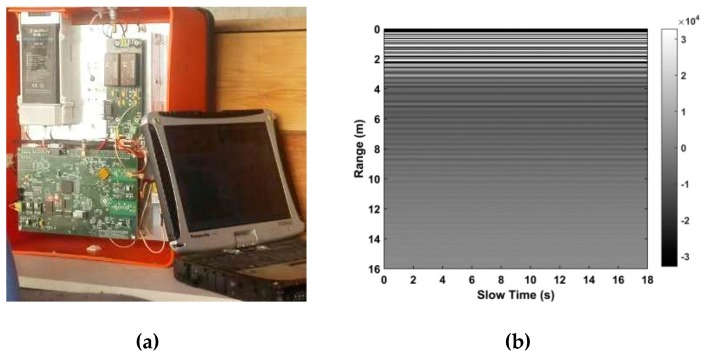
(**a**) The UWB radar and (**b**) the data for the first experiment with a human subject at a distance of 9 m from the radar.

**Figure 10 sensors-19-00095-f010:**
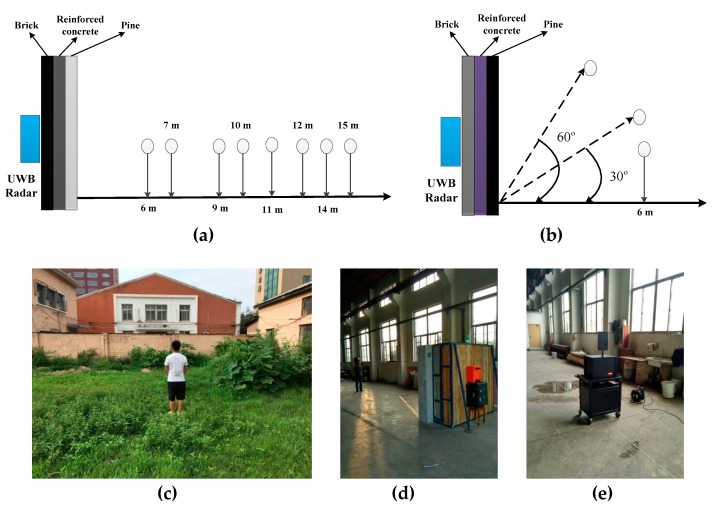
The experimental setup for (**a**) subjects in front of the radar, (**b**) subjects at an angle to the radar, (**c**) through-wall detection outdoors, (**d**) through-wall detection indoors and (**e**) the actuator.

**Figure 11 sensors-19-00095-f011:**
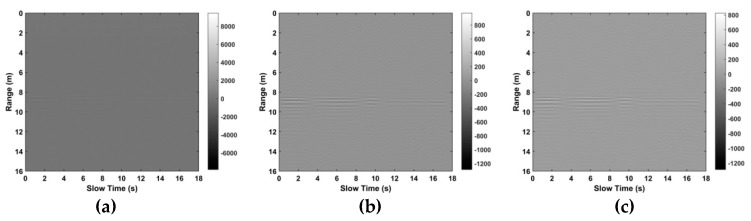
The results for the first experiment after (**a**) linear trend suppression (LTS), (**b**) range (fast time) filtering and (**c**) slow time filtering.

**Figure 12 sensors-19-00095-f012:**
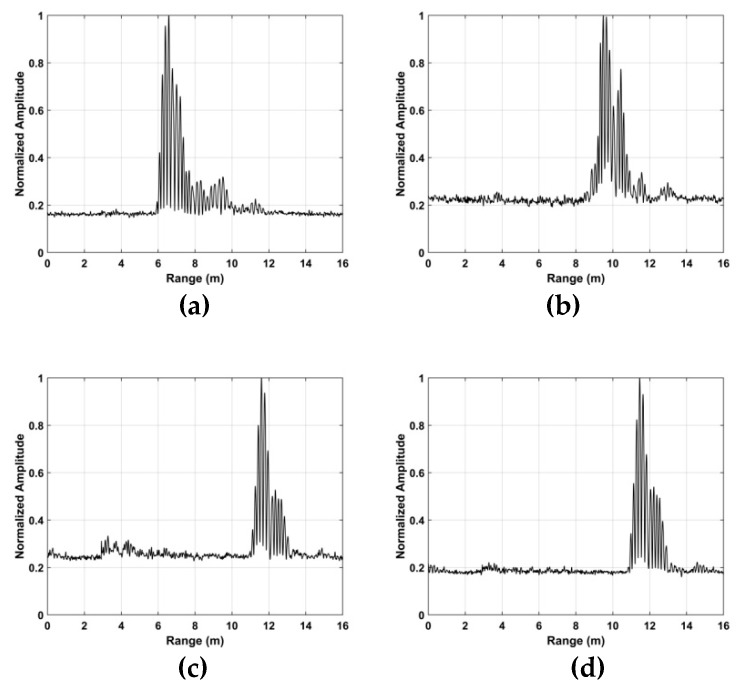
The KSD for the first experiment at distances of (**a**) 6 m, (**b**) 9 m, (**c**) 11 m and (**d**) 14 m from the radar.

**Figure 13 sensors-19-00095-f013:**
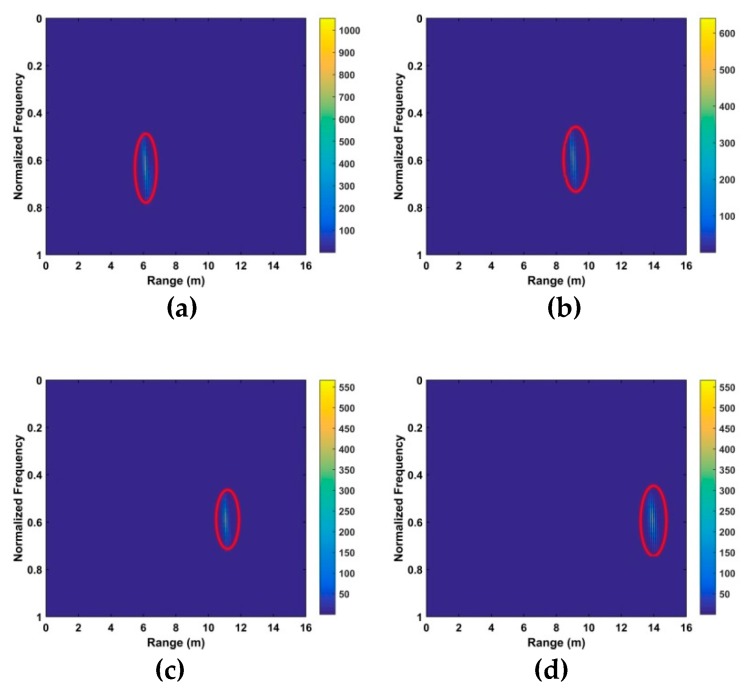
Range estimation for the first experiment at distances of (**a**) 6 m, (**b**) 9 m, (**c**) 11 m and (**d**) 14 m from the radar.

**Figure 14 sensors-19-00095-f014:**
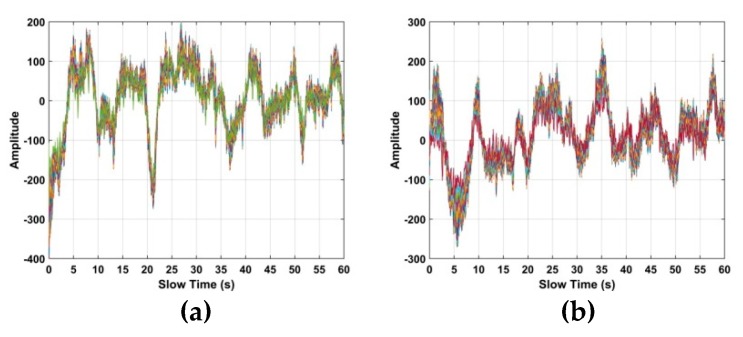
The signal in the ROI for the first experiment at distances of (**a**) 6 m, (**b**) 9 m, (**c**) 11 m and (**d**) 14 m from the radar.

**Figure 15 sensors-19-00095-f015:**
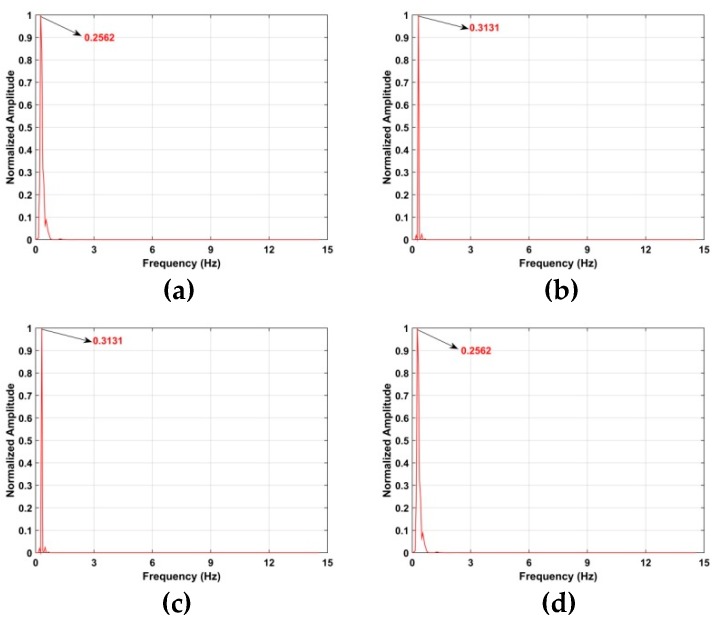
Frequency estimation for the first experiment at distances of (**a**) 6 m, (**b**) 9 m, (**c**) 11 m and (**d**) 14 m from the radar.

**Figure 16 sensors-19-00095-f016:**

Results using the CFAR method for subject III in the first experiment at distances of (**a**) 6 m, (**b**) 9 m and (**c**) 11 m from the radar.

**Figure 17 sensors-19-00095-f017:**
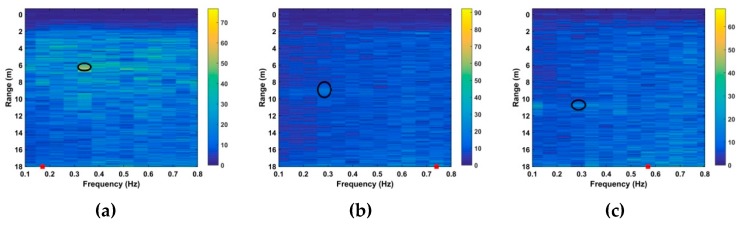
Results using AM for subject III in the first experiment at distances of (**a**) 6 m, (**b**) 9 m and (**c**) 11 m from the radar.

**Figure 18 sensors-19-00095-f018:**
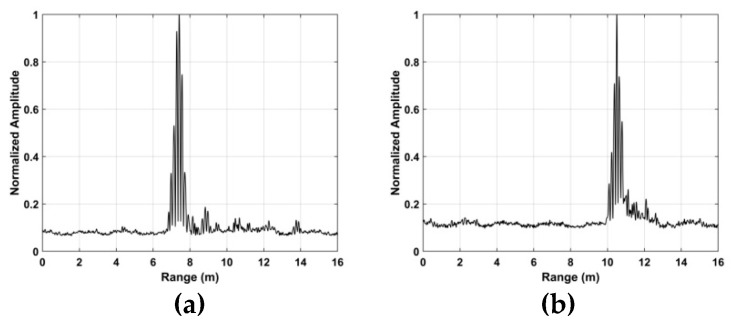
The KSD for the second experiment at distances of (**a**) 7 m, **(b**) 10 m, (**c**) 12 m and (**d**) 15 m from the radar.

**Figure 19 sensors-19-00095-f019:**
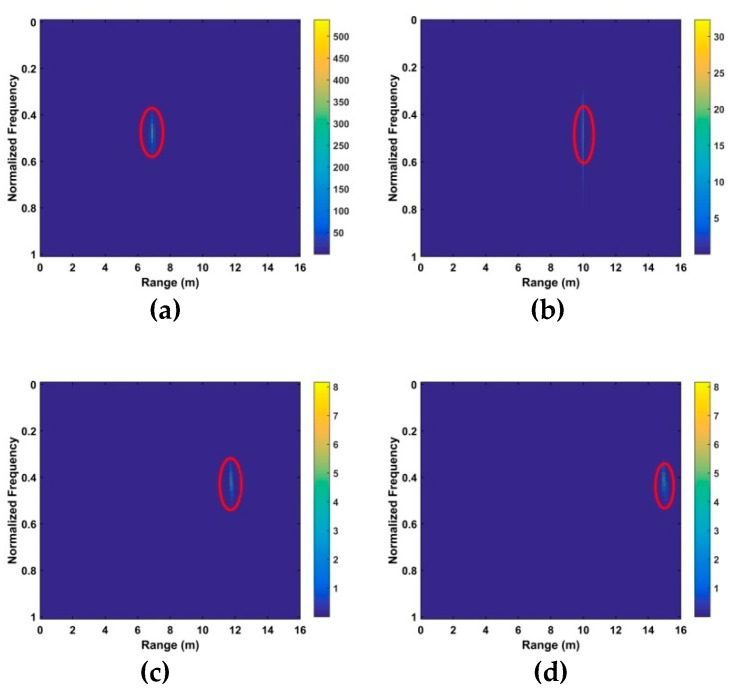
Range estimation for the second experiment at distances of (**a**) 7 m, (**b**) 10 m, (**c**) 12 m and (**d**) 15 m from the radar.

**Figure 20 sensors-19-00095-f020:**
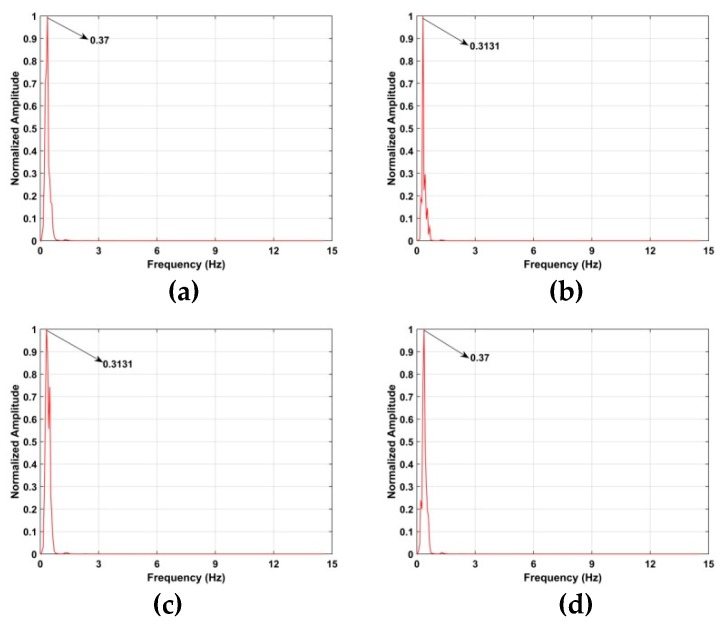
Frequency estimation for the second experiment at distances of (**a**) 7 m, (**b**) 10 m, (**c**) 12 m and (**d**) 15 m from the radar.

**Figure 21 sensors-19-00095-f021:**
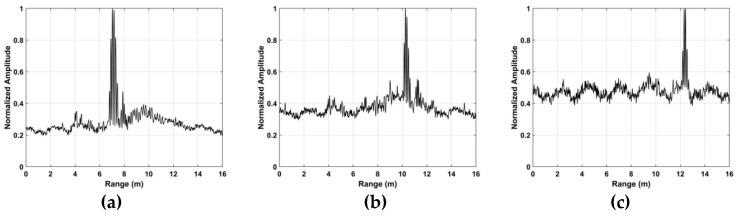
The KSD for the third experiment at distances of (**a**) 7 m, (**b**) 10 m and (**c**) 12 m from the radar.

**Figure 22 sensors-19-00095-f022:**
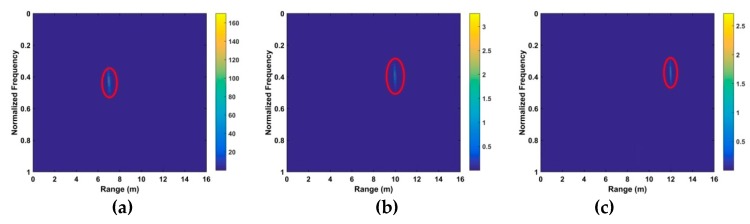
Third experiment range estimation at distances of (**a**) 7 m, (**b**) 10 m and (**c**) 12 m from the radar.

**Figure 23 sensors-19-00095-f023:**
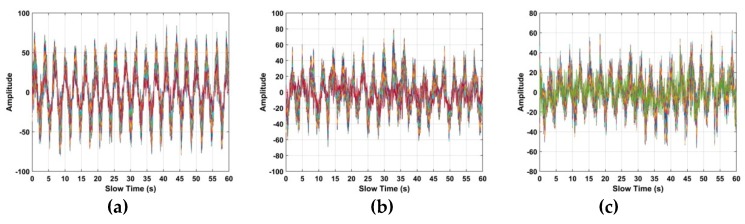
The signal in the ROI for the third experiment at distances of (**a**) 7 m, (**b**) 10 m and (**c**) 12 m from the radar.

**Figure 24 sensors-19-00095-f024:**
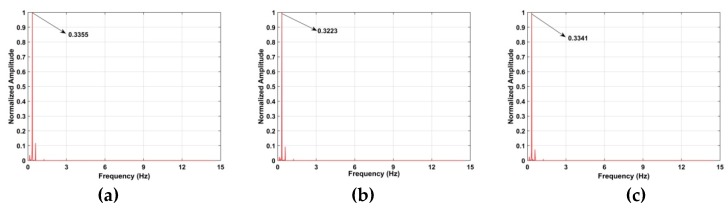
Third experiment frequency estimation at distances of (**a**) 7 m, (**b**) 10 m and (**c**) 12 m from the radar.

**Figure 25 sensors-19-00095-f025:**
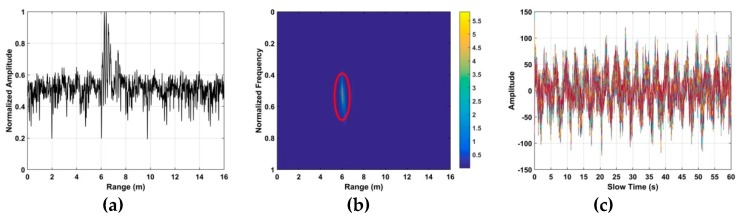
Results for the fourth experiment with the actuator outdoors (**a**) KSD, (**b**) range estimation and (**c**) the signal in the ROI.

**Figure 26 sensors-19-00095-f026:**
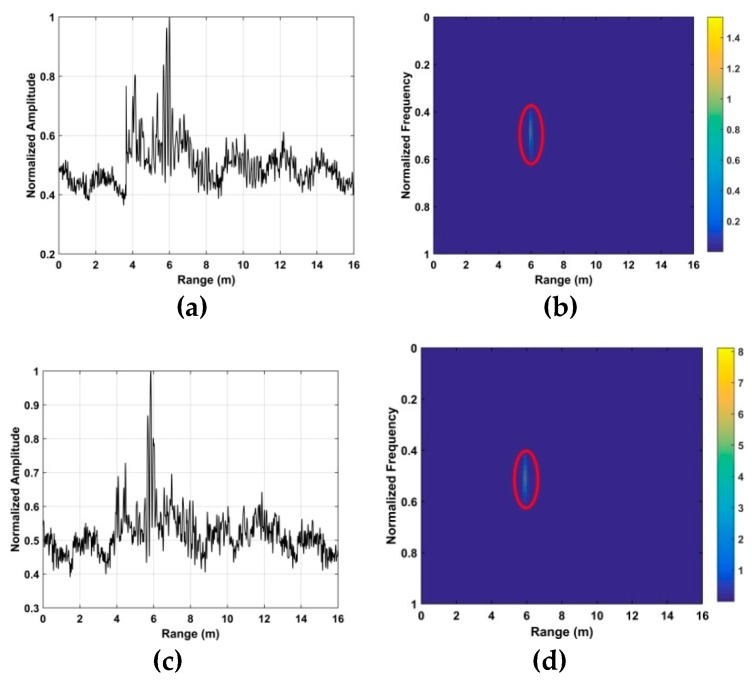
The results for the fifth experiment (**a**) KSD at 30°, (**b**) range at 30°, (**c**) KSD at 60° and (**d**) range at 60°.

**Figure 27 sensors-19-00095-f027:**
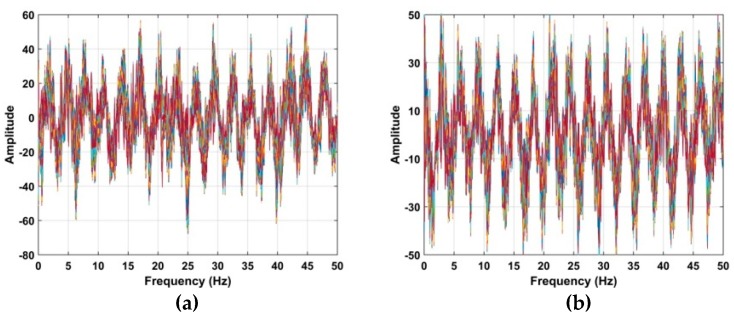
The signals in the ROI for the fifth experiment at angles of (**a**) 30° and (**b**) 60°.

**Table 1 sensors-19-00095-t001:** The UMB Impulse Radar Parameters.

Parameter	Value
center frequency	400 MHz
transmitted signal amplitude	50 V
pulse repeat frequency	600 KHz
number of averaged values (*N_A_*)	30
time window	124 ns
number of samples (*M*)	4092
input bandwidth of the Analog to Digital Converter (ADC)	2.3 GHz
ADC sampling rate	500 MHz
ADC sample size	12 bits
receiver dynamic range	72 dB

**Table 2 sensors-19-00095-t002:** VS Estimates for Four Subjects at Different Distances.

Subject	Gender	Height (cm)	Weight (kg)	Parameter	6 m	9 m	11 m
I	Female	158	48	Frequency (Hz)	0.26	0.31	0.31
				SNR (dB)	−4.92	−7.56	−8.29
II	Female	163	54	Frequency (Hz)	0.31	0.31	0.26
				SNR (dB)	−7.08	−10.6	−11.0
III	Male	178	84	Frequency (Hz)	0.37	0.31	0.37
				SNR (dB)	−6.52	−9.52	−12.9
IV	Male	182	76	Frequency (Hz)	0.37	0.31	0.37
				SNR (dB)	−7.12	−9.48	−11.3

**Table 3 sensors-19-00095-t003:** Performance for Subject I with Four Methods.

Method	Parameter	6 m	9 m	11 m
**CFAR**	**Range Error (m)**	**4.36**	**6.72**	**9.54**
	Frequency (Hz)	0.10	0.72	0.46
	SNR (dB)	−8.22	−12.86	−15.26
**Proposed**	**Range Error (m)**	**0.12**	**0.17**	**0.11**
	Frequency (Hz)	0.25	0.31	0.31
	SNR (dB)	−4.91	−7.55	−8.28
**MHOC**	**Range Error (m)**	**2.43**	**1.56**	**7.25**
	Frequency (Hz)	0.45	0.52	0.44
	SNR (dB)	−6.85	−9.58	−12.35
**AM**	**Range Error (m)**	**5.46**	**4.67**	**3.98**
	Frequency (Hz)	0.12	0.74	0.63
	SNR (dB)	0.84	−3.69	−6.59

**Table 4 sensors-19-00095-t004:** VS Estimates for Three Distances.

Subject	Gender	Height (cm)	Method	Parameter	7 m	10 m	12 m
**I**	Female	158	Proposed	Frequency (Hz)	0.37	0.31	0.31
				Range Error (m)	0.04	0.05	0.08
				SNR (dB)	−4.87	−6.76	−10.4
**II**	Female	163	FFT+ Window	Frequency (Hz)	0.14	0.20	0.20
				Range Error (m)	0.15	0.27	11.7
				SNR (dB)	−9.08	−13.7	−15.6
**IV**	Male	182	FFT	Frequency (Hz)	11.7	11.7	11.7
				Range Error (m)	6.70	9.70	11.7
				SNR (dB)	−29.4	−31.9	−32.2

**Table 5 sensors-19-00095-t005:** Results with Four Methods at Different Distances.

Method	Parameter	7 m	10 m	12 m
Proposed	Frequency (Hz)	0.35	0.32	0.33
	Deviation	0.66%	0.33%	0.24%
	Range Error (m)	0.026	0.043	0.040
FFT+Window	Frequency (Hz)	0.37	0.37	0.12
	Deviation	11%	11%	64%
	Range Error (m)	0.23	0.27	11.9
AM	Frequency (Hz)	0.34	0.34	0.34
	Deviation	2.5 %	2.5 %	2.5
	Range Error (m)	0.40	0.37	8.70
CFAR	Frequency (Hz)	0.37	0.37	0.43
	Deviation	11%	11%	28%
	Range Error (m)	0.30	0.32	11.7
MHOC	Frequency (Hz)	0.11	0.11	0.08
	Deviation	65%	65%	73%
	Range Error (m)	0.47	0.62	0.47

**Table 6 sensors-19-00095-t006:** Results for Different Azimuth Angles.

Method	Parameter	30°	60°
MHOC	Frequency (Hz)	0.37	0.26
	Deviation	11%	23%
	Range Error (m)	0.51	5.27
AM	Frequency (Hz)	0.34	0.74
	Deviation	2.5%	122%
	Range Error (m)	13.73	13.73
FFT + Window	Frequency (Hz)	0.14	0.14
	Deviation	57%	57%
	Range Error (m)	5.56	2.81
Proposed	Frequency (Hz)	0.37	0.37
	Deviation	11%	11%
	Range Error (m)	0.10	0.10

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
