# Peer review of "Random-Noise Denoising and Clutter Elimination of Human Respiration Movements Based on an Improved Time Window Selection Algorithm Using Wavelet Transform"

_sensors, 2018, doi:10.3390/s19010095_

Round 1
Reviewer 1 Report
This paper presented an improved sensing algorithm for reducing the effect of random noises and stationary clutters. Although the advantages of the proposed algorithm are clear when compared to other authors' works, the sensitivity of the used UWB radar with the proposed algorithm still cannot detect heartbeat frequency. Therefore, I don't this work is the state of the art. I will suggest the authors to reference the following paper, which detected heartbeat frequency using UWB radar with a phase-based algorithm.
L. Ren et al., “Phase-based methods for heart rate detection using UWB impulse Doppler radar,” IEEE Trans. Microw. Theory Techn., vol. 64, no. 10, pp. 3319-3331, Oct. 2016.
Author Response
We would like to thank you for the time spent in evaluating our submission. Your comments have helped tremendously in improving this paper.
L. Ren et al., “Phase-based methods for heart rate detection using UWB impulse Doppler radar,” IEEE Trans. Microw. Theory Techn., vol. 64, no. 10, pp. 3319-3331, Oct. 2016.
The reference has been added in this paper.
Reviewer 2 Report
Critical comments: 1. In the experimental section, please show the state of the human subject. Was the subject completely stationary during the experiment or having slight movements?
2. In the result section in table 3, there is a comparison of different algorithms with the proposed algorithm, however, there is no reference measurement. I’m curious as how the authors have compared the techniques.
3. In table 4, please indicate the unit of the range error. If the measurement unit is “meters”, then can you make an estimate that how much range error will lead to completely false reading for respiration frequency? Because I can see some range error as high as 11(meter?). Does it mean that the target is not properly localized? And so the breathing rate is completely incorrect for such huge range errors?
4. Lines 243-245: “This figure shows that d6 is concentrated in the 0.1 to 0.5 Hz frequency range, while the other wavelets are concentrated in higher (>1 Hz) or lower (<0.1 Hz) frequencies”. The figure 7, however, shows that the wavelets d1-d8 are all concentrated at same location. It would be better to show a zoomed view of each figure in order to visualize the difference as the author has claimed.
5. In the fifth experiment an actuator was used. Please explain more about the actuator.
6. Is the ROI constant for all the human subjects or it depends on the height, weight, size of the person? Please explain it in the paper.
7. I would suggest to include the following recent papers in the VS through IR-UWB radar in the references and show how your methods outperforms the results of this paper.
[1] Khan, Faheem, and Sung Ho Cho. "A detailed algorithm for vital sign monitoring of a stationary/non-stationary human through ir-uwb radar." Sensors 17.2 (2017): 290.
[2] Leem, Seong Kyu, Faheem Khan, and Sung Ho Cho. "Vital Sign Monitoring and Mobile Phone Usage Detection Using IR-UWB Radar for Intended Use in Car Crash Prevention." Sensors 17.6 (2017): 1240.
Minor mistakes:
1. Line 221. “samples are used to reduce”
2. In table 4, the fifth column has different order for the parameters for different subjects. Please use the same order for parameters such as Frequency à Range error à SNR for all the subjects. 3.In table 5, the range error for AM is mentioned as “error” only.
Author Response
We would like to thank you for the time spent in evaluating our submission. The comments have helped tremendously in improving this paper. Below you will find our point-by-point responses to your comments/questions.Critical comments:
1. In the experimental section, please show the state of the human subject. Was the subject completely stationary during the experiment or having slight movements?
response: The human subjects faced the radar breathing normally and kept still.
2. In the result section in table 3, there is a comparison of different algorithms with the proposed algorithm, however, there is no reference measurement. I’m curious as how the authors have compared the techniques.
response: In table 3, three well-known algorithms including the CFAR method [1], AM method [2], and HOC method are used as references to validate the proposed method. The compared results including the error between the true range and range estimate, SNR, and human respiratory frequency.
The proposed algorithm can provide the range estimates with the highest accuracy and highest SNR value among these four methods.
To validate the accuracy of frequency estimate, the actuator is used in the following text as shown in Table 5 and Table 6.
[1] Xu, Y.; et al. A novel method for automatic detection of trapped victims by ultrawideband radar, IEEE Trans. Geosci. Remote Sens., 2012, 50 (8): 3132-3142.
[2] Wu, S.; Tan, K.; Xia, Z.; Chen, J. Improved human respiration detection method via ultra-wideband radar in through-wall or other similar conditions, IET Radar Sonar Navig., 2016, 10 (3): 468-476.
[3] Xu, Y.; Wu, S.; Chen, C.; Chen, J.; Fang, G. Vital sign detection method based on multiple higher order cumulant for ultra-wideband radar, IEEE Trans. Geosci. Remote Sens. 2012, 50 (4): 1254-1265.
3. In table 4, please indicate the unit of the range error. If the measurement unit is “meters”, then can you make an estimate that how much range error will lead to completely false reading for respiration frequency? Because I can see some range error as high as 11(meter?). Does it mean that the target is not properly localized? And so the breathing rate is completely incorrect for such huge range errors?
reponse: As shown in Table 4, the unit for the range error is meter.
In this paper, the range is estimated using the SD-based kurtosis analysis.
Based on the range estimate, the ROI is defined, which can be used to estimate the breathing rate. Thus, the accurate range estimate is the key to estimate the breathing rate. The ROI cannot be obtained when the range is acquired with a larger error, which means the target cannot be properly localized. Thus, the breathing rate is completely incorrect for such huge range errors.
4. Lines 243-245: “This figure shows that d6 is concentrated in the 0.1 to 0.5 Hz frequency range, while the other wavelets are concentrated in higher (>1 Hz) or lower (<0.1 Hz) frequencies”. The figure 7, however, shows that the wavelets d1-d8 are all concentrated at same location. It would be better to show a zoomed view of each figure in order to visualize the difference as the author has claimed.
response: Figure 7 has been updated.
5. In the fifth experiment an actuator was used. Please explain more about the actuator.
reponse: The actuator is a metal plate, which is of size 0.25´0.3 m2 to imitate human respiration. The actuator moved at an amplitude of 3 mm and a frequency of 0.3333 Hz.
6. Is the ROI constant for all the human subjects or it depends on the height, weight, size of the person? Please explain it in the paper.
response: The ROI is independent of the height, weight, size of the person, and it constant for all the human subjects. The ROI is defined based on the distance estimate. Using (29), the index of the distance estimate can be acquired. The human respiration frequency is usually between 0.2 and 0.4 Hz with amplitude 0.5 to 1.5 cm. Thus, using the distance estimate (acquired accurately), a range consists of all pulses modulated by human respiratory can be determined. Here the range is considered as the region of interest (ROI) containing the respiration signal.
7. I would suggest to include the following recent papers in the VS through IR-UWB radar in the references and show how your methods outperforms the results of this paper.
[1] Khan, Faheem, and Sung Ho Cho. "A detailed algorithm for vital sign monitoring of a stationary/non-stationary human through ir-uwb radar." Sensors 17.2 (2017): 290.
[2] Leem, Seong Kyu, Faheem Khan, and Sung Ho Cho. "Vital Sign Monitoring and Mobile Phone Usage Detection Using IR-UWB Radar for Intended Use in Car Crash Prevention." Sensors 17.6 (2017): 1240.
response: These references have been added in this paper.
Minor mistakes:
8. Line 221. “samples are used to reduce”
response: Line 221 has been amended.
9. In table 4, the fifth column has different order for the parameters for different subjects. Please use the same order for parameters such as Frequency à Range error à SNR for all the subjects.
response: Table 4 has been amended.
10.In table 5, the range error for AM is mentioned as “error” only.
response: Table 5 has been amended.
Reviewer 3 Report
The authors propose a method for denoising and clutter elimination in the processing of impulse
ultra-wide band signals. Despite being mathematically sound (in the sense that the equations appear to be correct, no sure), I am not convinced that the novelty and the significances. The authors summarized their contributions in page 2 (line 62-line 71). However, I think those 5 points are the procedures of the proposed method. The main contribution might be in two folds, (1) improve the SNR based on one bandpass filter; (2) improve the system efficiency based on the selection of ROI. I suggest the authors clarify the novelty in the abstract and the introduction. The authors spend too much space on how to get the signals. In addition, I have the following concerns,
1) The authors should clarify which kind of vital signals to estimate in the title, abstract and conclusion of the paper.
2) The authors should check through the paper carefully. There are so many symbols in the paper. The authors should explain clearly what each symbol means. For instance, what does the ‘r’ and ‘p’ mean in Eq.2? In addition, the author employ ‘e’ to represent the convolution. I guess people always use ‘*’ to represent the convolution operation. What does the ‘*’ represent in Eq.5? The authors also use upper ‘*’ to denote complex conjugate in Eq. 25.
3) What is the SNR of the Figure 2(b)?
4) How to detect whether there is a subject or no subject (such as in Figure 5)? How about the situations with more than one subjects?
5) Why the authors select the d6 to estimate the respiration frequency?
6) How the ground truth is collected during the comparison between different methods?
Author Response
We would like to thank you for the time spent in evaluating our submission. The comments have helped tremendously in improving this paper. Below you will find our point-by-point responses to the comments/questions.
The authors propose a method for denoising and clutter elimination in the processing of impulse ultra-wide band signals. Despite being mathematically sound (in the sense that the equations appear to be correct, no sure), I am not convinced that the novelty and the significances. The authors summarized their contributions in page 2 (line 62-line 71). However, I think those 5 points are the procedures of the proposed method. The main contribution might be in two folds, (1) improve the SNR based on one bandpass filter; (2) improve the system efficiency based on the selection of ROI. I suggest the authors clarify the novelty in the abstract and the introduction. The authors spend too much space on how to get the signals. In addition, I have the following concerns, Response: The abstract and the introduction have been updated.
1) The authors should clarify which kind of vital signals to estimate in the title, abstract and conclusion of the paper.
Response: In this paper, the vital sign denotes human respiration movement, which has been given in the title, abstract and conclusion of the paper
2) The authors should check through the paper carefully. There are so many symbols in the paper. The authors should explain clearly what each symbol means. For instance, what does the ‘r’ and ‘p’ mean in Eq.2? In addition, the author employ ‘e’ to represent the convolution. I guess people always use ‘*’ to represent the convolution operation. What does the ‘*’ represent in Eq.5? The authors also use upper ‘*’ to denote complex conjugate in Eq. 25.
Response: All these errors have been amended in this paper.
3) What is the SNR of the Figure 2(b)?
Rersponse: The signal with AWGN is at an SNR of -10 dB.
4) How to detect whether there is a subject or no subject (such as in Figure 5)? How about the situations with more than one subjects?
Response: Based on the results as shown in Figure 5, 14, 20, and 23, the amplitudes of the signals in the time-frequency matrix are less than 2 when there is no subject in the detection environments. However, the amplitudes of the signals are up to several hundred whether in long range and short range using the developed radar. Thus, based on the amplitude of the acquired signals can determine whether there is a subject or no subject.
5) Why the authors select the d6 to estimate the respiration frequency?
Response: The frequency of human respiratory movements is usually in the range of 0.2-0.4 Hz. The signals reconstructed using d6 is concentrated in the 0.1 to 0.6 Hz frequency range, which meets the requirements of human respiratory movements. And the other wavelets are concentrated in higher (>1 Hz) or lower (<0.1 Hz) frequencies. As a result, to reduce noise the VS signals are extracted using d6.
6) How the ground truth is collected during the comparison between different methods?
Response: The actuator is of size 0.25´0.3 m2 to imitate human respiration.
The actuator moved at an amplitude of 3 mm and a frequency of 0.3333 Hz.
These ground truth are used to validate the proposed algorithm compared with different methods.
Round 2
Reviewer 1 Report
The author revised the manuscript nicely.
Reviewer 2 Report
Thanks for implementing the suggested changes in your paper and clarifying your work.
Reviewer 3 Report
The authors have address my concerns and the paper can be accepted for publication.